# Bayesian Inference of Individualized Treatment Effects using Multi-task Gaussian Processes

**Ahmed M. Alaa**
Electrical Engineering Department
University of California, Los Angeles
ahmedmalaa@ucla.edu

**Mihaela van der Schaar**
Department of Engineering Science
University of Oxford
mihaela.vanderschaar@eng.ox.ac.uk

## Abstract

Predicated on the increasing abundance of electronic health records, we investigate the problem of inferring *individualized* treatment effects using observational data. Stemming from the potential outcomes model, we propose a novel *multi-task* learning framework in which factual and counterfactual outcomes are modeled as the outputs of a function in a vector-valued reproducing kernel Hilbert space (vvRKHS). We develop a nonparametric Bayesian method for learning the treatment effects using a multi-task Gaussian process (GP) with a linear coregionalization kernel as a prior over the vvRKHS. The Bayesian approach allows us to compute individualized measures of confidence in our estimates via pointwise credible intervals, which are crucial for realizing the full potential of precision medicine. The impact of selection bias is alleviated via a *risk-based empirical Bayes* method for adapting the multi-task GP prior, which jointly minimizes the empirical error in factual outcomes and the uncertainty in (unobserved) counterfactual outcomes. We conduct experiments on observational datasets for an interventional social program applied to premature infants, and a left ventricular assist device applied to cardiac patients wait-listed for a heart transplant. In both experiments, we show that our method significantly outperforms the state-of-the-art.

## 1 Introduction

Clinical trials entail enormous costs: the average costs of multi-phase trials in vital therapeutic areas such as the respiratory system, anesthesia and oncology are $115.3 million, $105.4 million, and $78.6 million, respectively [1]. Moreover, due to the difficulty of patient recruitment, randomized controlled trials often exhibit small sample sizes, which hinders the discovery of heterogeneous therapeutic effects across different patient subgroups [2]. Observational studies are cheaper and quicker alternatives to clinical trials [3, 4]. With the advent of electronic health records (EHRs), currently deployed in more than 75% of hospitals in the U.S. according to the latest ONC data brief[1], there is a growing interest in using machine learning to infer heterogeneous treatment effects from readily available observational data in EHRs. This interest glints in recent initiatives such as STRATOS [3], which focuses on guiding observational medical research, in addition to various recent works on causal inference from observational data developed by the machine learning community [4-11].

Motivated by the plethora of EHR data and the potentiality of *precision medicine*, we address the problem of estimating *individualized* treatment effects (i.e. causal inference) using observational data. The problem differs from standard supervised learning in that for every subject in an observational cohort, we only observe the "factual" outcome for a specific treatment assignment, but never observe the corresponding "counterfactual" outcome[2], without which we can never know the true

treatment effect [4-9]. Selection bias creates a discrepancy in the feature distributions for the treated and control patient groups, which makes the problem even harder. Much of the classical works have focused on the simpler problem of estimating *average* treatment effects via unbiased estimators based on propensity score weighting (see [14] and the references therein). More recent works learn individualized treatment effects via regression models that view the subjects' treatment assignments as input features [4-13]. We provide a thorough review on these works in Section 3.

**Contribution** At the heart of this paper lies a novel conception of causal inference as a multi-task learning problem. That is, we view a subject's potential outcomes as the outputs of a vector-valued function in a reproducing kernel Hilbert space (vvRKHS) [15]. We propose a Bayesian approach for learning the treatment effects through a multi-task Gaussian process (GP) prior over the populations' potential outcomes. The Bayesian perspective on the multi-task learning problem allows reasoning about the unobserved counterfactual outcomes, giving rise to a loss function that quantifies the Bayesian risk of the estimated treatment effects while taking into account the uncertainty in counterfactual outcomes without explicit propensity modeling. Furthermore, we show that optimizing the multi-task GP hyper-parameters via *risk-based empirical Bayes* [16] is equivalent to minimizing the empirical error in the factual outcomes, with a regularizer that is proportional to the posterior uncertainty (variance) in counterfactual outcomes. We provide a feature space interpretation of our method showing its relation to previous works on domain adaptation [6, 8], empirical risk minimization [13], and tree-based learning [4, 5, 7, 9].

The Bayesian approach allows us to compute individualized measures of confidence in our estimates via pointwise credible intervals. With the exception of [5] and [9], all previous works do not associate their estimates with confidence measures, which hinders their applicability in formal medical research. While Bayesian credible sets do not guarantee frequentist coverage, recent results on the "honesty" (i.e. frequentist coverage) of adaptive credible sets in nonparametric regression may extend to our setting [16]. In particular, [Theorem 1, 16] shows that –under some extrapolation conditions– adapting a GP prior via risk-based empirical Bayes guarantees honest credible sets: investigating the validity of these results in our setting is an interesting topic for future research.

## 2 Problem Setup

We consider the setting in which a specific treatment is applied to a population of subjects, where each subject $i$ possesses a $d$-dimensional *feature* $X_i \in \mathcal{X}$, and two (random) *potential outcomes* $Y_i^{(1)}, Y_i^{(0)} \in \mathbb{R}$ that are drawn from a distribution $(Y_i^{(1)}, Y_i^{(0)})|X_i = x \sim \mathbb{P}(.|X_i = x)$, and correspond to the subject's response with and without the treatment, respectively. The realized causal effect of the treatment on subject $i$ manifests through the random variable $(Y_i^{(1)} - Y_i^{(0)}) \,|\, X_i = x$. Hence, we define the *individualized treatment effect* (ITE) for subjects with a feature $X_i = x$ as

$$T(x) = \mathbb{E}\left[ Y_i^{(1)} - Y_i^{(0)} \,\middle|\, X_i = x \right]. \tag{1}$$

Our goal is to conduct the *causal inference* task of estimating the function $T(x)$ from an *observational* dataset $\mathcal{D}$, which typically comprises $n$ independent samples of the random tuple $\{X_i, W_i, Y_i^{(W_i)}\}$, where $W_i \in \{0, 1\}$ is a treatment assignment indicator that indicates whether or not subject $i$ has received the treatment under consideration. The outcomes $Y_i^{(W_i)}$ and $Y_i^{(1-W_i)}$ are known as the *factual* and the *counterfactual* outcomes, respectively [6, 9]. Treatment assignments are generally dependent on features, i.e. $W_i \not\perp X_i$. The conditional distribution $\mathbb{P}(W_i = 1|X_i = x)$, also known as the *propensity score* of subject $i$ [13, 14], reflects the underlying policy for assigning the treatment to subjects. Throughout this paper, we respect the standard assumptions of *unconfoundedness* (or *ignorability*) and *overlap*: this setting is known in the literature as the "potential outcomes model with unconfoundedness" [4-11].

Individual-based causal inference using observational data is challenging. Since we only observe one of the potential outcomes for every subject $i$, we never observe the treatment effect $Y_i^{(1)} - Y_i^{(0)}$ for any of the subjects, and hence we cannot resort to standard supervised learning to estimate $T(x)$. Moreover, the dataset $\mathcal{D}$ exhibits *selection bias*, which may render the estimates of $T(x)$ inaccurate if the treatment assignment for individuals with $X_i = x$ is strongly biased (i.e. $\mathbb{P}(W_i = 1|X_i = x)$ is close to 0 or 1). Since our primary motivation for addressing this problem comes from its application potential in precision medicine, it is important to associate our estimate of $T(.)$ with a pointwise measure of confidence in order to properly guide therapeutic decisions for individual patients.

# 3 Multi-task Learning for Causal Inference

**Vector-valued Potential Outcomes Function** We adopt the following *signal-in-white-noise* model for the potential outcomes:

$$Y_i^{(w)} = f_w(X_i) + \epsilon_{i,w}, \ w \in \{0,1\}, \tag{2}$$

where $\epsilon_{i,w} \sim \mathcal{N}(0, \sigma_w^2)$ is a Gaussian noise variable. It follows from (2) that $\mathbb{E}[Y_i^{(w)} \mid X_i = x] = f_w(x)$, and hence the ITE can be estimated as $\hat{T}(x) = \hat{f}_1(x) - \hat{f}_0(x)$. Most previous works that estimate $T(x)$ via *direct modeling* learn a single-output regression model that treats the treatment assignment as an input feature, i.e. $f_w(x) = f(x, w), f(., .) : \mathcal{X} \times \{0, 1\} \to \mathbb{R}$, and estimate the ITE as $\hat{T}(x) = \hat{f}(x, 1) - \hat{f}(x, 0)$ [5-9]. We take a different perspective by introducing a new multi-output regression model comprising a *potential outcomes* (PO) function $\mathbf{f}(.) : \mathcal{X} \to \mathbb{R}^2$, with $d$ inputs (features) and 2 outputs (potential outcomes); the ITE estimate is the projection of the estimated PO function on the vector $\mathbf{e} = [-1 \ \ 1]^T$, i.e. $\hat{T}(x) = \hat{\mathbf{f}}^T(x) \mathbf{e}$.

Consistent pointwise estimation of the ITE function $T(x)$ requires restricting the PO function $\mathbf{f}(x)$ to a smooth function class [9]. To this end, we model the PO function $\mathbf{f}(x)$ as belonging to a *vector-valued Reproducing Kernel Hilbert Space* (vvRKHS) $\mathcal{H}_{\mathbf{K}}$ equipped with an inner product $\langle ., . \rangle_{\mathcal{H}_{\mathbf{K}}}$, and with a *reproducing kernel* $\mathbf{K} : \mathcal{X} \times \mathcal{X} \to \mathbb{R}^{2 \times 2}$, where $\mathbf{K}$ is a (symmetric) positive semi-definite matrix-valued function [15]. Our choice for the vvRKHS is motivated by its algorithmic advantages; by virtue of the *representer theorem*, we know that learning the PO function entails estimating a finite number of coefficients evaluated at the input points $\{X_i\}_{i=1}^n$ [17].

**Multi-task Learning** The vector-valued model for the PO function conceptualizes causal inference as a multi-task learning problem. That is, $\mathcal{D} = \{X_i, W_i, Y_i^{(W_i)}\}_{i=1}^n$ can be thought of as comprising training data for two learning tasks with target functions $f_0(.)$ and $f_1(.)$, with $W_i$ acting as the "task index" for the $i^{th}$ training point [15]. For an estimated PO function $\hat{\mathbf{f}}(x)$, the true loss functional is

$$\mathcal{L}(\hat{\mathbf{f}}) = \int_{x \in \mathcal{X}} \left( \hat{\mathbf{f}}^T(x) \mathbf{e} - T(x) \right)^2 \cdot \mathbb{P}(X = x) \, dx. \tag{3}$$

The loss functional in (3) is known as the *precision in estimating heterogeneous effects* (PEHE), and is commonly used to quantify the "goodness" of $\hat{T}(x)$ as an estimate of $T(x)$ [4-6, 8]. A conspicuous challenge that arises when learning the "PEHE-optimal" PO function $\mathbf{f}$ is that we cannot compute the empirical PEHE for a particular $\mathbf{f} \in \mathcal{H}_{\mathbf{K}}$ since the treatment effect samples $\{Y_i^{(1)} - Y_i^{(0)}\}_{i=1}^n$ are not available in $\mathcal{D}$. On the other hand, using a loss function that evaluates the losses of $\hat{f}_0(x)$ and $\hat{f}_1(x)$ separately (as in conventional multi-task learning [Sec. 3.2, 15]) can be highly problematic: in the presence of a strong selection bias, the empirical loss for $\mathbf{f}(.)$ with respect to factual outcomes may not generalize to counterfactual outcomes, leading to a large PEHE loss. In order to gain insight into the structure of the optimal PO function, we consider an "oracle" that has access to counterfactual outcomes. For such an oracle, the finite-sample empirical PEHE is

$$\hat{\mathcal{L}}(\hat{\mathbf{f}}; \mathbf{K}, \mathbf{Y}^{(\mathbf{W})}, \mathbf{Y}^{(\mathbf{1} - \mathbf{W})}) = \frac{1}{n} \sum_{i=1}^n \left( \hat{\mathbf{f}}^T(X_i) \mathbf{e} - (1 - 2W_i) \left( Y_i^{(1 - W_i)} - Y_i^{(W_i)} \right) \right)^2, \tag{4}$$

where $\mathbf{Y}^{(\mathbf{W})} = [Y_i^{(W_i)}]_i$ and $\mathbf{Y}^{(\mathbf{1} - \mathbf{W})} = [Y_i^{(1 - W_i)}]_i$. When $\mathbf{Y}^{(\mathbf{1} - \mathbf{W})}$ is accessible, the PEHE-optimal PO function $\mathbf{f}(.)$ is given by the following representer Theorem.

**Theorem 1** (Representer Theorem for Oracle Causal Inference). *For any $\hat{\mathbf{f}}^* \in \mathcal{H}_{\mathbf{K}}$ satisfying*

$$\hat{\mathbf{f}}^* = \arg \min_{\hat{\mathbf{f}} \in \mathcal{H}_{\mathbf{K}}} \hat{\mathcal{L}}(\hat{\mathbf{f}}; \mathbf{K}, \mathbf{Y}^{(\mathbf{W})}, \mathbf{Y}^{(\mathbf{1} - \mathbf{W})}) + \lambda ||\hat{\mathbf{f}}||_{\mathcal{H}_{\mathbf{K}}}^2, \ \lambda \in \mathbb{R}_+, \tag{5}$$

*we have that $\hat{T}^*(.) = \mathbf{e}^T \hat{\mathbf{f}}^*(.) \in span\{\tilde{\mathbf{K}}(., X_1), \ldots, \tilde{\mathbf{K}}(., X_n)\}$, where $\tilde{\mathbf{K}}(., .) = \mathbf{e}^T \mathbf{K}(., .) \mathbf{e}$. That is, $\hat{T}^*(.)$ admits a representation $\hat{T}^*(.) = \sum_{i=1}^n \alpha_i \tilde{\mathbf{K}}(., X_i)$, $\alpha = [\alpha_1, \ldots, \alpha_n]^T$, where*

$$\alpha = (\tilde{\mathbf{K}}(\mathbf{X}, \mathbf{X}) + n \lambda \mathbf{I})^{-1}((\mathbf{1} - 2\mathbf{W}) \odot (\mathbf{Y}^{(\mathbf{1} - \mathbf{W})} - \mathbf{Y}^{(\mathbf{W})})), \tag{6}$$

*where $\odot$ denotes component-wise product, $\tilde{\mathbf{K}}(\mathbf{X}, \mathbf{X}) = (\tilde{\mathbf{K}}(X_i, X_j))_{i,j}$, $\mathbf{W} = [W_1, \ldots, W_n]^T$.* $\square$

**A Bayesian Perspective** Theorem 1 follows directly from the generalized representer Theorem [17] (A proof is provided in [17]), and it implies that regularized empirical PEHE minimization in vvRKHS is equivalent to Bayesian inference with a Gaussian process (GP) prior [Sec. 2.2, 15]. Therefore, we can interpret $\hat{T}^*(.)$ as the posterior mean of $T(.)$ given a GP prior with a covariance kernel $\tilde{\mathbf{K}}$, i.e. $T \sim \mathcal{GP}(0, \tilde{\mathbf{K}})$. We know from Theorem 1 that $\tilde{\mathbf{K}} = \mathbf{e}^T \mathbf{K} \mathbf{e}$, hence the prior on $T(.)$ is equivalent to a *multi-task* GP prior on the PO function $\mathbf{f}(.)$ with a kernel $\mathbf{K}$, i.e. $\mathbf{f} \sim \mathcal{GP}(0, \mathbf{K})$.

The Bayesian view of the problem is advantageous for two reasons. First, as discussed earlier, it allows computing individualized (pointwise) measures of uncertainty in $\hat{T}(.)$ via posterior credible intervals. Second, it allows reasoning about the unobserved counterfactual outcomes in a Bayesian fashion, and hence provides a natural proxy for the oracle learner's empirical PEHE in (4). Let $\theta \in \Theta$ be a kernel *hyper-parameter* that parametrizes the multi-task GP kernel $\mathbf{K}_\theta$. We define the Bayesian PEHE risk $R(\theta, \hat{\mathbf{f}}; \mathcal{D})$ for a point estimate $\hat{\mathbf{f}}$ as follows

$$R(\theta, \hat{\mathbf{f}}; \mathcal{D}) = \mathbb{E}_\theta \left[ \hat{\mathcal{L}}(\hat{\mathbf{f}}; \mathbf{K}_\theta, \mathbf{Y}^{(\mathbf{W})}, \mathbf{Y}^{(\mathbf{1}-\mathbf{W})}) \,\middle|\, \mathcal{D} \right]. \tag{7}$$

The expectation in (7) is taken with respect to $\mathbf{Y}^{(\mathbf{1}-\mathbf{W})}|\mathcal{D}$. The Bayesian PEHE risk $R(\theta, \hat{\mathbf{f}}; \mathcal{D})$ is simply the oracle learner's empirical loss in (4) marginalized over the posterior distribution of the unobserved counterfactuals $\mathbf{Y}^{(\mathbf{1}-\mathbf{W})}$, and hence it incorporates the posterior uncertainty in counterfactual outcomes without explicit propensity modeling. The optimal hyper-parameter $\theta^*$ and interpolant $\hat{\mathbf{f}}^*(.)$ that minimize the Bayesian PEHE risk are given in the following Theorem.

**Theorem 2** (Risk-based Empirical Bayes). *The minimizer $(\hat{\mathbf{f}}^*, \theta^*)$ of $R(\theta, \hat{\mathbf{f}}; \mathcal{D})$ is given by*

$$\hat{\mathbf{f}}^* = \mathbb{E}_{\theta^*}[\mathbf{f}\,|\,\mathcal{D}], \quad \theta^* = \arg \min_{\theta \in \Theta} \left[ \underbrace{\left\| \mathbf{Y}^{(\mathbf{W})} - \mathbb{E}_\theta[\mathbf{f}\,|\,\mathcal{D}] \right\|_2^2}_{\text{Empirical factual error}} + \underbrace{\left\| \mathrm{Var}_\theta[\mathbf{Y}^{(\mathbf{1}-\mathbf{W})}\,|\,\mathcal{D}] \right\|_1}_{\text{Posterior counterfactual variance}} \right],$$

*where $\mathrm{Var}_\theta[.|.]$ is the posterior variance and $\|.\|_p$ is the $p$-norm.* $\square$

The proof is provided in Appendix A. Theorem 2 shows that hyper-parameter selection via *risk-based empirical Bayes* is instrumental in alleviating the impact of selection bias. This is because, as the Theorem states, $\theta^*$ minimizes the empirical loss of $\hat{\mathbf{f}}^*$ with respect to factual outcomes, and uses the posterior variance of the counterfactual outcomes as a regularizer. Hence, $\theta^*$ carves a kernel that not only fits factual outcomes, but also generalizes well to counterfactuals. It comes as no surprise that $\hat{\mathbf{f}}^* = \mathbb{E}_{\theta^*}[\mathbf{f}\,|\,\mathcal{D}]$; $\mathbb{E}_{\theta^*}[\mathbf{f}\,|\,\mathcal{D}, \mathbf{Y}^{(\mathbf{1}-\mathbf{W})}]$ is equivalent to the oracle's solution in Theorem 1, hence by the law of iterated expectations, $\mathbb{E}_{\theta^*}[\mathbf{f}\,|\,\mathcal{D}] = \mathbb{E}_{\theta^*}[\mathbb{E}_{\theta^*}[\mathbf{f}\,|\,\mathcal{D}, \mathbf{Y}^{(\mathbf{1}-\mathbf{W})}]\,|\,\mathcal{D}]$ is the oracle's solution marginalized over the posterior distribution of counterfactuals.

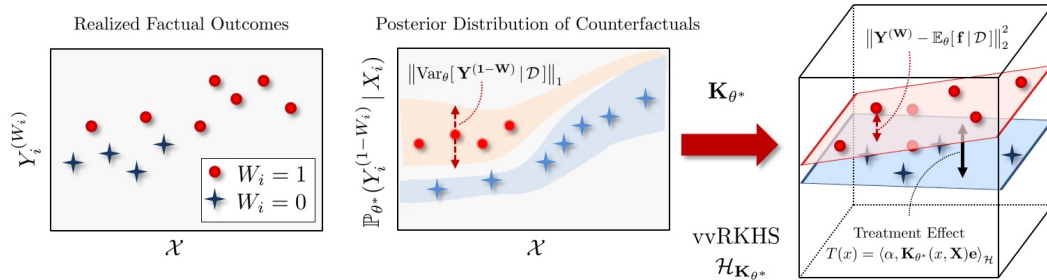

Figure 1: Pictorial depiction for model selection via risk-based empirical Bayes.

**Related Works** A feature space interpretation of Theorem 2 helps creating a conceptual equivalence between our method and previous works. For simplicity of exposition, consider a finite-dimensional vvRKHS in which the PO function resides: we can describe such a space in terms of a feature map $\mathbf{\Phi} : \mathcal{X} \to \mathbb{R}^p$, where $\mathbf{K}(x, x') = \langle \mathbf{\Phi}(x), \mathbf{\Phi}(x') \rangle$ [Sec. 2.3, 15]. Every PO function $\mathbf{f} \in \mathcal{H}_\mathbf{K}$ can be represented as $\mathbf{f} = \langle \alpha, \mathbf{\Phi}(x) \rangle$, and hence the two response surfaces $f_o(.)$ and $f_1(.)$

are represented as hyperplanes in the transformed feature space as depicted in Fig. 1 (right). The risk-based empirical Bayes method attempts to find a feature map $\mathbf{\Phi}$ and two hyperplanes that best fit the factual outcomes (right panel in Fig. 1) while minimizing the posterior variance in counterfactual outcomes (middle panel in Fig. 1). This conception is related to that of *counterfactual regression* [6, 8], which builds on ideas from co-variate shift and domain adaptation [19] in order to jointly learn a response function $f$ and a "balanced" representation $\mathbf{\Phi}$ that makes the distributions $\mathbb{P}(\mathbf{\Phi}(X_i = x)|W_i = 1)$ and $\mathbb{P}(\mathbf{\Phi}(X_i = x)|W_i = 0)$ similar. Our work differs from [6, 8] in the following aspects. First, our Bayesian multi-task formulation provides a direct estimate of the PEHE: (7) is an unbiased estimator of the finite-sample version of (3). Contrarily, [Eq. 2, 6] creates a coarse proxy for the PEHE by using the nearest-neighbor factual outcomes in replacement of counterfactuals, whereas [Eq. 3, 8] optimizes a generalization bound which may largely overestimate the true PEHE for particular hypothesis classes. [6] optimizes the algorithm's hyper-parameters by assuming (unrealistically) that counterfactuals are available in a held-out sample, whereas [8] uses an ad hoc nearest-neighbor approximation. Moreover, unlike the case in [6], our multi-task formulation protects the interactions between $W_i$ and $X_i$ from being lost in high-dimensional feature spaces.

Most of the previous works estimate the ITE via *co-variate adjustment* ($G$-computation formula) [4, 5, 7, 11, 20]; the most remarkable of these methods are the nonparametric *Bayesian additive regression trees* [5] and *causal forests* [4, 9]. We provide numerical comparisons with both methods in Section 5. [11] also uses Gaussian processes, but with the focus of modeling treatment response curves over time. Counterfactual risk minimization is another framework that is applicable only when the propensity score $\mathbb{P}(W_i = 1|X_i = x)$ is known [12, 13]. [25] uses deep networks to infer counterfactuals, but requires some of the data to be drawn from a randomized trial.

## 4 Causal Multi-task Gaussian Processes (CMGPs)

In this Section, we provide a recipe for Bayesian causal inference with the prior $\mathbf{f} \sim \mathcal{GP}(0, \mathbf{K}_\theta)$. We call this model a *Causal Multi-task Gaussian Process* (CMGP).

**Constructing the CMGP Kernel** As it is often the case in medical settings, the two response surfaces $f_0(.)$ and $f_1(.)$ may display different levels of heterogeneity (smoothness), and may have different relevant features. Standard intrinsic coregionalization models for constructing vector-valued kernels impose the same covariance parameters for all outputs [18], which limits the interaction between the treatment assignments and the patients' features. To that end, we construct a *linear model of coregionalization* (LMC) [15], which mixes two intrinsic coregionalization models as follows

$$\mathbf{K}_\theta(x, x') = \mathbf{A}_0 \, k_0(x, x') + \mathbf{A}_1 \, k_1(x, x'), \tag{8}$$

where $k_w(x, x'), w \in \{0, 1\}$, is the *radial basis function* (RBF) with automatic relevance determination, i.e. $k_w(x, x') = \exp\left(-\frac{1}{2}(x - x')^T \mathbf{R}_w^{-1} (x - x')\right)$, $\mathbf{R}_w = \text{diag}(\ell_{1,w}^2, \ell_{2,w}^2, \ldots, \ell_{d,w}^2)$, with $\ell_{d,w}$ being the *length scale* parameter of the $d^{th}$ feature in $k_w(., .)$, whereas $\mathbf{A}_0$ and $\mathbf{A}_1$ are given by

$$\mathbf{A}_0 = \begin{bmatrix} \beta_{00}^2 & \rho_0 \\ \rho_0 & \beta_{01}^2 \end{bmatrix}, \quad \mathbf{A}_1 = \begin{bmatrix} \beta_{10}^2 & \rho_1 \\ \rho_1 & \beta_{11}^2 \end{bmatrix}. \tag{9}$$

The parameters $(\beta_{ij}^2)_{ij}$ and $(\rho_i)_i$ determine the variances and correlations of the two response surfaces $f_0(x)$ and $f_1(x)$. The LMC kernel introduces degrees of freedom that allow the two response surfaces to have different covariance functions and relevant features. When $\beta_{00} >> \beta_{01}$ and $\beta_{11} >> \beta_{10}$, the length scale parameter $\ell_{d,w}$ can be interpreted as the relevance of the $d^{th}$ feature to the response surface $f_w(.)$. The set of all hyper-parameters is $\theta = (\sigma_0, \sigma_1, \mathbf{R}_0, \mathbf{R}_1, \mathbf{A}_0, \mathbf{A}_1)$.

**Adapting the Prior via Risk-based Empirical Bayes** In order to avoid overfitting to the factual outcomes $\mathbf{Y}^{(\mathbf{W})}$, we evaluate the empirical error in factual outcomes via leave-one-out cross-validation (LOO-CV) with Bayesian regularization [24]; the regularized objective function is thus given by $\hat{R}(\theta; \mathcal{D}) = \eta_0 \, Q(\theta) + \eta_1 \, \|\theta\|_2^2$, where

$$Q(\theta) = \left\| \text{Var}_\theta[\mathbf{Y}^{(\mathbf{1}-\mathbf{W})} \,|\, \mathcal{D}] \right\|_1 + \sum_{i=1}^{n} \left( Y_i^{(W_i)} - \mathbb{E}_\theta[\mathbf{f}(X_i) \,|\, \mathcal{D}_{-i}] \right)^2, \tag{10}$$

and $\mathcal{D}_{-i}$ is the dataset $\mathcal{D}$ with subject $i$ removed, whereas $\eta_0$ and $\eta_1$ are the Bayesian regularization parameters. For the second level of inference, we use the improper Jeffrey's prior as an ignorance

prior for the regularization parameters, i.e. $\mathbb{P}(\eta_0) \propto \frac{1}{\eta_0}$ and $\mathbb{P}(\eta_1) \propto \frac{1}{\eta_1}$. This allows us to integrate out the regularization parameters [Sec. 2.1, 24], leading to a revised objective function $\hat{R}(\theta; \mathcal{D}) = n \log(Q(\theta)) + (10 + 2\,d) \log(\|\theta\|_2^2)$ [Eq. (15), 24]. It is important to note that LOO-CV with squared loss has often been considered to be unfavorable in ordinary GP regression as it leaves one degree of freedom undetermined [Sec. 5.4.2, 5]; this problem does not arise in our setting since the term $\left\|\mathrm{Var}_\theta\big[\,\mathbf{Y}^{(\mathbf{1-W})}\,|\,\mathcal{D}\,\big]\right\|_1$ involves all the variance parameters, and hence the objective function $\hat{R}(\theta; \mathcal{D})$ does not depend solely on the posterior mean.

**Causal Inference via CMGPs** Algorithm 1 sums up the entire causal inference procedure. It first invokes the routine `Initialize-hyperparameters`, which uses the sample variance and up-crossing rate of $\mathbf{Y}^{(\mathbf{W})}$ to initialize $\theta$ (see Appendix B). Such an automated initialization procedure allows running our method without any user-defined inputs, which facilitates its usage by researchers conducting observational studies. Having initialized $\theta$ (line 3), the algorithm finds a locally optimal $\theta^*$ using gradient descent (lines 5-12), and then estimates the ITE function and the associated credible intervals (lines 13-17). ($\mathbf{X} = [\{X_i\}_{W_i=0}, \{X_i\}_{W_i=1}]^T$, $\mathbf{Y} = [\{Y_i^{(W_i)}\}_{W_i=0}, \{Y_i^{(W_i)}\}_{W_i=1}]^T$, $\mathbf{\Sigma} = \mathrm{diag}(\sigma_0^2\,\mathbf{I}_{n-n_1}, \sigma_1^2\,\mathbf{I}_{n_1})$, $n_1 = \sum_i W_i$, $\mathrm{erf}(x) = \frac{1}{\sqrt{\pi}} \int_{-x}^x e^{-y^2}\,dy$, and $\mathbf{K}_\theta(x) = (\mathbf{K}_\theta(x, X_i))_i$.)

We use a re-parametrized version of the Adaptive Moment Estimation (ADAM) gradient descent algorithm for optimizing $\theta$ [21]; we first apply a transformation $\phi = \exp(\theta)$ to ensure that all covariance parameters remain positive, and then run ADAM to minimize $\hat{R}(\log(\phi_t); \mathcal{D})$. The ITE function is estimated as the posterior mean of the CMGP (line 14). The credible interval $\mathcal{C}_\gamma(x)$ with a Bayesian coverage of $\gamma$ for a subject with feature $x$ is defined as $\mathbb{P}_\theta(T(x) \in \mathcal{C}_\gamma(x)) = \gamma$, and is computed straightforwardly using the error function of the normal distribution (lines 15-17). The computational burden of Algorithm 1 is dominated by the $O(n^3)$ matrix inversion in line 13; for large observational studies, this can be ameliorated using conventional sparse approximations [Sec. 8.4, 23].

---

**Algorithm 1** Causal Inference via CMGPs

1: **Input:** Observational dataset $\mathcal{D}$, Bayesian coverage $\gamma$
2: **Output:** ITE function $\hat{T}(x)$, credible intervals $\mathcal{C}_\gamma(x)$
3: $\theta \leftarrow$ `Initialize-hyperparameters`$(\mathcal{D})$
4: $\phi^0 \leftarrow \exp(\theta), t \leftarrow 0, m_t \leftarrow 0, v_t \leftarrow 0,$
5: **repeat**
6: $\quad m_{t+1} \leftarrow \beta_1\, m_t + (1 - \beta_1) \cdot \phi_t \odot \nabla_\phi \hat{R}(\log(\phi_t); \mathcal{D})$
7: $\quad v_{t+1} \leftarrow \beta_2\, v_t + (1 - \beta_2) \cdot (\phi_t \odot \nabla_\phi \hat{R}(\log(\phi_t); \mathcal{D}))^2$
8: $\quad \hat{m}_{t+1} \leftarrow m_t/(1 - \beta_1^t), \hat{v}_{t+1} \leftarrow v_t/(1 - \beta_2^t)$
9: $\quad \phi_{t+1} \leftarrow \phi_t \odot \exp\left(-\eta \cdot \hat{m}_{t+1}/(\sqrt{\hat{v}_{t+1}} + \epsilon)\right)$
10: $\quad t \leftarrow t + 1$
11: **until** convergence
12: $\theta^* \quad \leftarrow \log(\phi_{t-1})$
13: $\mathbf{\Lambda}_{\theta^*} \leftarrow (\mathbf{K}_{\theta^*}(\mathbf{X}, \mathbf{X}) + \mathbf{\Sigma})^{-1}$
14: $\hat{T}(x) \leftarrow (\mathbf{K}_{\theta^*}^T(x)\, \mathbf{\Lambda}_{\theta^*}\, \mathbf{Y})^T \mathbf{e}$
15: $\mathbf{V}(x) \leftarrow \mathbf{K}_{\theta^*}(x, x) - \mathbf{K}_{\theta^*}(x)\, \mathbf{\Lambda}_{\theta^*}\, \mathbf{K}_{\theta^*}^T(x)$
16: $\hat{I}(x) \leftarrow \mathrm{erf}^{-1}(\gamma)\, (2\mathbf{e}^T \mathbf{V}(x)\mathbf{e})^{\frac{1}{2}}$
17: $\mathcal{C}_\gamma(x) \leftarrow [\hat{T}(x) - \hat{I}(x), \hat{T}(x) + \hat{I}(x)]$

---

# 5 Experiments

Since the ground truth counterfactual outcomes are never available in real-world observational datasets, evaluating causal inference algorithms is not straightforward. We follow the semi-synthetic experimental setup in [5, 6, 8], where covariates and treatment assignments are real but outcomes are simulated. Experiments are conducted using the IHDP dataset introduced in [5]. We also introduce a new experimental setup using the UNOS dataset: an observational dataset involving end-stage cardiovascular patients wait-listed for heart transplantation. Finally, we illustrate the clinical utility and significance of our algorithm by applying it to the real outcomes in the UNOS dataset.

**The IHDP dataset** The Infant Health and Development Program (IHDP) is intended to enhance the cognitive and health status of low birth weight, premature infants through pediatric follow-ups and parent support groups [5]. The semi-simulated dataset in [5, 6, 8] is based on covariates from a real randomized experiment that evaluated the impact of the IHDP on the subjects' IQ scores at the age of three: selection bias is introduced by removing a subset of the treated population. All outcomes (response surfaces) are simulated. The response surface data generation process was not designed to favor our method: we used the standard non-linear "Response Surface B" setting in [5]

(also used in [6] and [8]). The dataset comprises 747 subjects (608 control and 139 treated), and there are 25 covariates associated with each subject.

**The UNOS dataset**[3]  The United Network for Organ Sharing (UNOS) dataset contains information on every heart transplantation event in the U.S. since 1987. The dataset also contains information on patients registered in the heart transplantation wait-list over the years, including those who died before undergoing a transplant. Left Ventricular Assistance Devices (LVADs) were introduced in 2001 as a life-saving therapy for patients awaiting a heart donor [26]; the survival benefits of LVADs are very heterogeneous across the patients' population, and it is unclear to practitioners how outcomes vary across patient subgroups. It is important to learn the heterogeneous survival benefits of LVADs in order to appropriately re-design the current transplant priority allocation scheme [26].

We extracted a cohort of patients enrolled in the wait-list in 2010; we chose this year since by that time the current continuous-flow LVAD technology became dominant in practice, and patients have been followed up sufficiently long to assess their survival. (Details of data processing is provided in Appendix C.) After excluding pediatric patients, the cohort comprised 1,006 patients (774 control and 232 treated), and there were 14 covariates associated with each patient. The outcomes (survival times) generation model is described as follows: $\sigma_0 = \sigma_1 = 1$, $f_0(x) = \exp((x + \frac{1}{2})\Omega)$, and $f_1(x) = \Omega\, x - \omega$, where $\Omega$ is a random vector of regression coefficients sampled uniformly from $[0, 0.1, 0.2, 0.3, 0.4]$, and $\omega$ is selected for a given $\Omega$ so as to adjust the average survival benefit to 5 years. In order to increase the selection bias, we estimate the propensity score $\mathbb{P}(W_i = 1 | X_i = x)$ using logistic-regression, and then, sequentially, with probability 0.5 we remove the control patient whose propensity score is closest to 1, and with probability 0.5 we remove a random control patient. A total of 200 patients are removed, leading to a cohort with 806 patients. The resulting dataset is more biased than IHDP, and hence poses a greater inferential challenge.

Table 1: Results on the IHDP and UNOS datasets (lower $\sqrt{\text{PEHE}}$ is better).

| | | IHDP | | UNOS | |
|---|---|---|---|---|---|
| | | In-sample $\sqrt{\text{PEHE}}$ | Out-of-sample $\sqrt{\text{PEHE}}$ | In-sample $\sqrt{\text{PEHE}}$ | Out-of-sample $\sqrt{\text{PEHE}}$ |
| ♡ | **CMGP** | **0.59 ± 0.01** | **0.76 ± 0.01** | **1.7 ± 0.10** | **1.8 ± 0.13** |
| | GP | 2.1 ± 0.11 | 2.3 ± 0.14 | 4.1 ± 0.15 | 4.5 ± 0.20 |
| ♣ | BART | 2.0 ± 0.13 | 2.2 ± 0.17 | 3.5 ± 0.17 | 3.9 ± 0.23 |
| | CF | 2.4 ± 0.21 | 2.8 ± 0.23 | 3.8 ± 0.25 | 4.3 ± 0.31 |
| | VTRF | 1.4 ± 0.07 | 2.2 ± 0.16 | 4.5 ± 0.35 | 4.9 ± 0.41 |
| | CFRF | 2.7 ± 0.24 | 2.9 ± 0.25 | 4.7 ± 0.21 | 5.2 ± 0.32 |
| ♠ | BLR | 5.9 ± 0.31 | 6.1 ± 0.41 | 5.7 ± 0.21 | 6.2 ± 0.30 |
| | BNN | 2.1 ± 0.11 | 2.2 ± 0.13 | 3.2 ± 0.10 | 3.3 ± 0.12 |
| | CFRW | 1.0 ± 0.07 | 1.2 ± 0.08 | 2.7 ± 0.07 | 2.9 ± 0.11 |
| ★ | $k$NN | 3.2 ± 0.12 | 4.2 ± 0.22 | 5.2 ± 0.11 | 5.4 ± 0.12 |
| | PSM | 4.9 ± 0.31 | 4.9 ± 0.31 | 4.6 ± 0.12 | 4.8 ± 0.16 |
| ♢ | TML | 5.2 ± 0.35 | 5.2 ± 0.35 | 6.2 ± 0.31 | 6.2 ± 0.31 |

**Benchmarks**  We compare our algorithm with: ♣ Tree-based methods (BART [5], causal forests (CF) [4, 9], virtual-twin random forests (VTRF) [7], and counterfactual random forests (CFRF) [7]), ♠ Balancing counterfactual regression (Balancing linear regression (BLR) [6], balancing neural networks (BNN) [6], and counterfactual regression with Wasserstein distance metric (CFRW) [8]), ★ Propensity-based and matching methods ($k$ nearest-neighbor ($k$NN), propensity score matching (PSM)), ♢ Doubly-robust methods (Targeted maximum likelihood (TML) [22]), and ♡ Gaussian process-based methods (separate GP regression for treated and control with marginal likelihood maximization (GP)). Details of all these benchmarks are provided in Appendix D.

Following [4-9], we evaluate the performance of all algorithms by reporting the square-root of $\text{PEHE} = \frac{1}{n} \sum_{i=1}^{n} ((f_1(X_i) - f_0(X_i)) - \mathbb{E}[Y_i^{(1)} - Y_i^{(0)} | X_i = x])^2$, where $f_1(X_i) - f_0(X_i)$ is

the estimated treatment effect. We evaluate the PEHE via a Monte Carlo simulation with 1000 realizations of both the IHDP and UNOS datasets, where in each experiment we run all the benchmarks with 60/20/20 train-validation-test splits. Counterfactuals are never made available to any of the benchmarks. We run Algorithm 1 with the a learning rate of 0.01 and with the standard setting prescribed in [21] (i.e. $\beta_1 = 0.9, \beta_2 = 0.999, \epsilon = 10^{-8}$). We report both the in-sample and out-of-sample PEHE estimates: the former corresponds to the accuracy of the estimated ITE in a retrospective cohort study, whereas the latter corresponds to the performance of a clinical decision support system that provides out-of-sample patients with ITE estimates [8]. The in-sample PEHE metrics is non-trivial since we never observe counterfactuals even in the training phase.

**Results** As can be seen in Table 1, CMGPs outperform all other benchmarks in terms of the PEHE in both the IHDP and UNOS datasets. The benefit of the risk-based empirical Bayes method manifest in the comparison with ordinary GP regression that fits the treated and control populations by evidence maximization. The performance gain of CMGPs with respect to GPs increase in the UNOS dataset as it exhibits a larger selection bias, hence naïve GP regression tends to fit a function to the factual outcomes that does not generalize well to counterfactuals. Our algorithm is also performing better than all other nonparametric tree-based algorithms. In comparison to BART, our algorithm places an adaptive prior on a smooth function space, and hence it is capable of achieving faster posterior contraction rates than BART, which places a prior on a space of discontinuous functions [16]. Similar insights apply to the frequentist random forest algorithms. CMGPs also outperform the different variants of counterfactual regression in both datasets, though CFRW is competitive in the IHDP experiment. BLR performs badly in both datasets as it balances the distributions of the treated and control populations by variable selection, and hence it throws away informative features for the sake of balancing the selection bias. The performance gain of CMGPs with respect to BNN and CFRW shows that the multi-task learning framework is advantageous: through the linear coregionalization kernel, CMGPs preserves the interactions between $W_i$ and $X_i$, and hence is capable of capturing highly non-linear (heterogeneous) response surfaces.

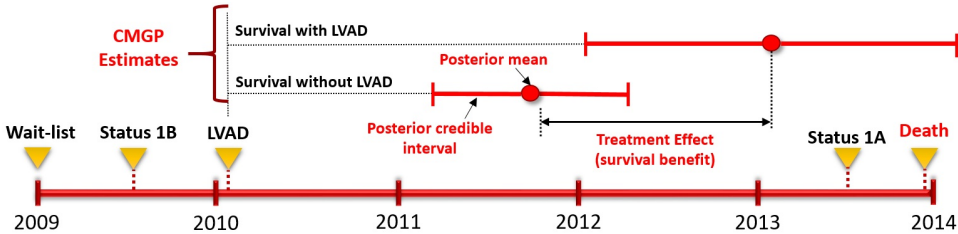

Figure 2: Pathway for a representative patient in the UNOS dataset.

## 6 Discussion: Towards Precision Medicine

To provide insights into the clinical utility of CMGPs, we ran our algorithm on all patients in the UNOS dataset who were wait-listed in the period 2005-2010, and used the real patient survival times as outcomes. The current transplant priority allocation scheme relies on a coarse categorization of patients that does not take into account their individual risks; for instance, all patients who have an LVAD are thought of as benefiting from it equally. We found a substantial evidence in the data that this leads to wrong clinical decision. In particular, we found that 10.3% of wait-list patients for whom an LVAD was implanted exhibit a delayed assignment to a high priority allocation in the wait-list. One of such patients has her pathway depicted in Fig. 2: she was assigned a high priority (status 1A) in June 2013, but died shortly after, before her turn to get a heart transplant. Her late assignment to the high priority status was caused by an overestimated benefit of the LVAD she got implanted in 2010; that is, the wait-list allocation scheme assumed she will attain the "populational average" survival benefit from the LVAD. Our algorithm had a much more conservative estimate of her survival; since she was diabetic, her individual benefit from the LVAD was less than the populational average. We envision a new priority allocation scheme in which our algorithm is used to allocate priorities based on the individual risks in a personalized manner.

## Footnotes

[1]https://www.healthit.gov/sites/default/files/briefs/

[2]Some works refer to this setting as the "logged bandits with feedback" [12, 13].

[3]https://www.unos.org/data/

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
