[Reviews · NeurIPS 2017]

Reviewer 1



The authors propose a method of estimating treatment effectiveness T(x) from a vector of patient features x. Treatment effectiveness is defined as (health outcome with treatment Yw) - (health outcome without treatment Y(1-w)). Presumably a health outcome might be something like survival time. If a patient survives 27 months with the treatment and only 9 without then the effectiveness T(x) would be 18 months? The authors estimate models of "outcome with treatment" and "outcome without treatment" jointly using RKHS kernel approximations on the whole dataset (I think there is a shared kernel). For a specific patient the effectiveness is based on the actual outcome of the patient which will be based on their features and their treatment condition minus the population model for the features of the opposite or counterfactual treatment condition. The population model is represented by an RKHS approximation which generates Bayesian posterior confidence bounds allowing the algorithm to integrate over uncertainty in the treatment effectiveness. I like that the authors call this out in line 139: integrating over the uncertainty in the counterfactual condition improves their results. They employ an empirical Bayes framework in which they optimize for hyperparameters that control distributions over health outcomes. They use a loss function that tries to maximize accuracy of the actual outcome observed for a patient with specific features and minimizing the variance of the counterfactual ... This seemed a bit counterintuitive to me - wouldn't you want to maximize the variance of the counterfactual in order to be consistent with the principle of least commitment / maximum entropy? Don't make conclusions that are stronger than the data supports?? They end up using a mixture of two kernels to provide additional flexibility in modeling. The algorithm does a matrix inversion which I think is common in RKHS style approximation which may limit its scalability somewhat (also true of Gaussian Process based approaches). The example they show has only 1000 patients which is very small for an EHR dataset generally which can have millions of patients. The goal of the authors is to be able to handle situations with strong selection bias (that is, the treatment is correlated with the patient features so we don't get an unbiased selection of treatments for any given condition). They end up having to filter the datasets they work with to create stronger selection bias. It is not clear whether this was a necessary step for them to demonstrate advantage over the other algorithms they benchmark against. For their filtered dataset they show metrics with around 1/3 the error rate of competing state of the art methods. They attribute the gains to the ability of their algorithm to compensate for selection bias by integrating over uncertainty in the counterfactual??? The paper has a lot of technical jargon which is not obviously necessary to explaining the intuitions clearly. Do we really need the details of the representer theorem in the paper? I would prefer using the space to describe intuitions on why it works more clearly. line 27: "glints" is used as a verb in an awkward way I liked Figure 1, but it took me a while to understand it. It might be worth explicitly calling out the roles of the 3 subfigures in the caption. If you used fewer data points it might be more obvious that the realized factual outcomes and the counterfatual distributions are supposed to align ... maybe it would make sense to stack these graphs vertically so they share the same x axis? line 53: "helps creating" is an awkward expression lines 251 ... it would be nice if the authors could describe a bit more about the competing bench mark methods and why they perform the way they do? There is a short related work section but it doesn't use any of the names of the algorithms. I don't know if these are the same as the comparison benchmarks or not. Would the ordering be the same with a dataset that has low selection bias?

Reviewer 2



In this paper, the authors proposed a novel multi-task learning framework using a nonparametric Bayesian method. Also, they demonstrate that the proposed method significantly outperforms the state-of-the-art. This reviewer has the following comments: - Bayesian perspective on the multi-task learning is not a novel concept. The authors need to review other literatures and include them in the introduction. - It is mentioned that the computational burden of algorithm 1 is dominated by the O(n^3) matrix inversion in line 13. It would be interesting to compare to the other methods in terms of the computational efficiency. Also, the author mentioned that for large observational studies, it can be ameliorated using sparse approximations. But it would be interesting to see the robustness and computational efficiency for large dataset.

Reviewer 3



The authors develop a nonparametric bayesian method in order to infer individualised treatment effects on observational data. In particular they use a multi task gaussian process prior with a linear coregionalization kernel as covariance. They use a regularized risk-based empirical Bayes method to find the optimal set of hyperparameters and interpolant. It jointly minimises the error of factual outcomes and the posterior counterfactual variance. They show that their method outperforms the state of the art in two observational datasets. Significance I believe that this is an interesting model that infers ITEs on observational data. Clarity Some parts of the model proposed by the authors are well-described and the structure of the paper makes it relatively easy to follow. However, some sections would benefit from clearer explanations: 1. the paragraph entitled "Related works". The explanation seems to be too concise and sometimes imprecise: for example, the authors do not define \alpha (in the text nor in the Figure), the reference is not accurate: Sec 3.2 [15] instead of Sec 2.3 [15], the response surface f_0 and f_1 are not mathematically defined. As a consequence I believe that the feature based interpretation lacks clarity although it is one of the main contribution of the paper. 2. the choice of the linear model of regularisation could be justified more precisely, for example the choice and influence of the parameters in A_0 and A_1 (equation (9)). In the experiment section, the Bayesian coverage value \gamma is not given. In the IHDP and the UNOS datasets, could the authors explain how the selection biases that are introduced affect the model? In particular, the selection bias that is introduced in the IHDP dataset is not described. Correctness Equation (10): it seems that Q(\theta) still depends on the whole dataset through the first term (posterior counterfactual variance) although the authors are evaluating the error via LOO-CV. In the proof in the supplementary, indexes 'i' are missing on many of the differentials.